# Positive and negative mood states do not influence cross-modal auditory distraction in the serial-recall paradigm

**Saskia Kaiser**[ID]*, **Axel Buchner, Raoul Bell**[ID]

Department of Experimental Psychology, Heinrich Heine University Düsseldorf, Düsseldorf, Germany

* saskia.kaiser@hhu.de

## Abstract

The aim of this study was to examine whether positive and negative mood states affect auditory distraction in a serial-recall task. The duplex-mechanism account differentiates two types of auditory distraction. The changing-state effect is postulated to be rooted in interference-by-process and to be automatic. The auditory-deviant effect is attributed to attentional capture by the deviant distractors. Only the auditory-deviant effect, but not the changing-state effect, should be influenced by emotional mood states according to the duplex-mechanism account. Four experiments were conducted to test how auditory distraction is affected by emotional mood states. Mood was induced by autobiographical recall (Experiments 1 and 2) or the presentation of emotional pictures (Experiments 3 and 4). Even though the manipulations were successful in inducing changes in mood, neither positive mood (Experiments 1 and 3) nor negative mood (Experiments 2 and 4) had any effect on distraction despite large samples sizes ($N = 851$ in total). The results thus are not in line with the hypothesis that auditory distraction is affected by changes in mood state. The results support an automatic-capture account according to which the auditory-deviant effect and the changing-state effect are mainly stimulus-driven effects that are rooted in the automatic processing of the to-be-ignored auditory stream.

**Data Availability Statement:** The datasets of the experiments as well as the supplementary material are available in the Open Science Framework repository at https://osf.io/tqjwr/.

## Introduction

When auditory distraction is studied in the lab, emotional states are often seen as an extraneous influence on performance that has to be controlled. In consequence, researchers often try to create emotionally neutral settings in laboratory experiments and the analysis of auditory distraction focuses only on the cognitive aspects of performance (for reviews, see [1, 2]). This contrasts with our everyday experience in which we are rarely ever in a completely neutral mood, but may feel sad, happy, aroused, or threatened. For example, students may have to ignore auditory distractors when taking a fear-inducing exam; workers in open-plan offices may have to maintain their concentration in the face of background noises on exciting and unpleasant workdays alike. This raises the question of whether results on auditory distraction obtained in highly controlled, emotionally neutral settings can be generalized to emotionally

**Funding:** The publication fee is funded by the open access fund of Heinrich Heine University Düsseldorf. The research (the four studies presented in the manuscript) did not receive external funding.

**Competing interests:** The authors have declared that no competing interests exist.

loaded situations in which auditory distractors have to be ignored in negative or positive mood states.

Even though auditory distraction has been conceptualized as being primarily determined by properties of the to-be-ignored information (e.g., the degree to which it deviates from a previous train of stimuli; see, for example, [3, 4]), it seems possible that auditory distraction is determined by emotional state. It has often been postulated that distraction by auditory stimuli is not only detrimental, but may serve an important adaptive function [e.g., 5–7]. Specifically, auditory signals such as alarms and human speech can be of relevance for the individual even if the auditory modality is nominally irrelevant to the ongoing task. Therefore, it seems maladaptive to completely stop the processing of the auditory input. Some degree of processing of the nominally irrelevant channel may be necessary to determine whether or not stimuli outside the focus of attention are of relevance for higher-order goals. However, given that demanding tasks are disrupted by devoting some degree of processing resources to the task-irrelevant modality, the attentional system has the delicate task of balancing out the conflicting goals of protecting ongoing processes from interference while ensuring the openness of the system to signals of higher-order relevance [5]. It is at least conceivable that the balance of these conflicting goals is largely affected by emotional factors which would imply that mood states are of central importance for understanding auditory distraction. Alternatively, it is also possible that auditory stimuli disrupt performance in a primarily stimulus-driven manner in which case the mood state of the individual should be of little importance [e.g., 8]. The aim of the present study is to test the effect of mood on auditory distraction in four well-powered studies, relying on effective mood-induction procedures and the well-established serial-recall paradigm to assess the behavioural effects of auditory distraction on working memory.

The serial-recall paradigm [9–11] is one of the best-established paradigms to measure auditory distraction in the lab. The immediate recall of information is severely disrupted when auditory distractors have to be ignored. This phenomenon is referred to as the irrelevant-sound effect. In the serial-recall paradigm, a list of visual targets is presented sequentially to participants who have to recall the targets immediately after presentation or following a short retention interval. In irrelevant-sound experiments, participants have to ignore auditory distractors while rehearsing the targets. Three types of auditory distractors are often distinguished from each other. Steady-state sequences consist of a single auditory distractor which is repeatedly presented (e.g., A A A A A A A A). Auditory-deviant sequences are similar but contain a distractor (the deviant distractor) that differs from the rest of the distractors in the to-be-ignored sequence (e.g., A A A A A B A A). The auditory-deviant effect [12–14] refers to the observation that auditory-deviant sequences are more disruptive than control sequences that do not contain a deviant stimulus. Changing-state sequences consist of distractor items that differ from each other (e.g., A B C D E F G H). The changing-state effect [15, 16] describes the observation that changing-state sequences disrupt performance more than steady-state sequences. Both the auditory-deviant effect and the changing-state effect are considered benchmarks of working memory [17] which underlines their importance for theories of working memory.

The influence of emotional factors on auditory distraction in the serial-recall paradigm has received surprisingly little attention. Exceptions are studies examining the effect of distractor words with emotional meaning in comparison to neutral distractor words, showing a higher distraction by emotionally loaded—especially negative—distractors compared to neutral distractors [18–21]. However, we know of no study investigating the influences of emotional mood states of the participants on basic forms of auditory distraction such as the auditory-deviant effect and the changing-state effect. Examining such influences is interesting because the duplex-mechanism account [22–24] postulates that the auditory-deviant effect and the

changing-state effect arise from completely different mechanisms, only one of which should be affected by emotional mood states. According to this account, phenomena of auditory distraction can be classified into two different types, one of which is automatic while the other depends on attentional control. The changing-state effect is a prototypical example of automatic interference-by-process. More precisely, according to the duplex-mechanism account, once changes between consecutive distractors are detected, order information is automatically extracted from the auditory stream. The automatic processing of order is assumed to occur for changing-state sequences but not for steady-state sequences. The pre-attentional processing of the order of the distractor items interferes with the voluntary processing of the order of the target items required by the serial-recall task and thus interferes with the rehearsal of the to-be-remembered target sequence. According to the duplex-mechanism account, the processing of the task-irrelevant order is automatic in the sense that it occurs independently of global attentional modulation. Therefore, the changing-state effect is postulated to remain unaffected by the individual's emotional-motivational state [22].

The auditory-deviant effect, by contrast, is due to attentional capture according to the duplex-mechanism account [22–24]. It is assumed that the deviant distractor violates the expectation built up by the repeating previous distractor stimuli and thus disengages attention from the targets. Within the duplex-mechanism account, attentional capture by auditory-deviant sequences is defined as being susceptible to, and dependent on, the individual´s global attentional state. Therefore, this account predicts the auditory-deviant effect to be influenced by emotional-motivational factors that modulate the trade-off between the deployment of attention to the serial-recall task and the allocation of attention to the task-irrelevant modality. Specifically, the duplex-mechanism account implies that in the case of the auditory-deviant effect attentional "distraction (. . .) is not a function merely of the properties of the distracting material itself (. . .) but also factors internal to the individual (. . .). [T]here exists not only inter-person variation in the overall capacity for cognitive control through increased task engagement (. . .) but also intraindividual variation over time, which can be influenced by a range of factors including task demands, emotional state, and motivational factors" [22, p.33]. Given that the hypothesis of a differential influence of emotional states on interference-by-process and attention capture has not yet been tested, the present study was designed to test whether emotional states have the differential effects on the changing-state effect and the auditory-deviant effect that are predicted by the duplex-mechanism account.

The effects of mood states on distraction should depend on the respective mood state. Specifically, positive and negative moods should affect distraction differently. According to the most influential theory on the effect of emotional mood states on distraction, positive affect leads to a broadening of the attentional focus and thus increases distraction by irrelevant stimuli [e.g., 25–29] while negative affect leads to a narrowing of the attentional focus and thus decreases the influence of task-irrelevant stimuli on performance [e.g., 30–32], relative to control conditions with neutral mood. The broaden-and-build-theory [27, 33] focuses on the influence of positive affect on selective attention. Positive mood is assumed to enhance cognitive flexibility and to cause a propensity to explore and to take in new information [27, 28]. This implies that attention is less likely to stay closely focused on nominally task-relevant stimuli in elevated mood states [25, 34] which increases distraction by irrelevant stimuli [26, 35] due to a relaxation of inhibitory control [35]. Negative affective states, by contrast, are assumed to cause an increase of attentional control and a narrowing of the attentional focus [e.g., 30–32] which should improve performance in selective-attention tasks [30, 36].

Regarding cross-modal auditory distraction, however, the available empirical evidence does not uniformly support the aforementioned theories but is sparse and scattered. While there is a lack of studies examining the effects of mood on auditory distraction in the serial-recall

paradigm, some evidence is available for cross-modal attention capture in oddball paradigms where distraction is primarily measured in terms of an increase in response latencies in simple classification tasks rather than the proportion of correct responses. Pacheco-Unguetti and Parmentier [37] found that deviance distraction was more pronounced when participants were in a happy mood than when they were in a neutral mood. Contrary to the prediction that negative mood improves selective attention, Pacheco-Unguetti and Parmentier [38] found that distraction in response to auditory deviants was more pronounced in sad mood than in neutral mood. By contrast, Hoskin et al. [39] found that experimentally induced anxiety did not affect distraction by auditory deviants. Inconsistent results have also been obtained for psychophysiological correlates of attention switching. Some studies have reported that the P3 component of the event-related potential in response to distractor sounds—that is often thought to be associated with the orienting to task-irrelevant sounds [e.g., 40]—was reduced when negative or positive pictures were displayed in comparison to when neutral pictures were displayed [41, 42]. However, in other studies the P3 was enhanced in response to novel sounds when participants watched negative material [43–45]. Overall, the available body of evidence is currently inconclusive as to whether, and, if so, how positive and negative emotional states affect auditory distraction.

Here, we report four high-powered experiments to test the effects of positive (Experiments 1 and 3) and negative (Experiments 2 and 4) mood states on auditory distraction in the serial-recall paradigm. In Experiments 1 and 2, we used a combination of well-established mood-induction procedures, autobiographical recall and music, which have been shown to be very effective in inducing different mood states [e.g., 37, 38, 46–49]. If the trade-off between relevant and irrelevant information flexibly depends on emotional state, auditory distraction should be affected by the experimentally induced mood states. The aforementioned theory on the effect of mood states on distraction predicts that distraction should increase in positive mood states and decrease in negative mood states [e.g., 26, 35, 36]. The duplex-mechanism account of auditory distraction [22] makes the differential prediction that only the auditory-deviant effect should depend on the individuals' emotional states while the changing-state effect should occur as an automatic consequence of the perceptual processing of auditory changes. An alternative view is that auditory distraction is a primarily stimulus-driven process. Specifically, it has been proposed that the detection of changes or unexpected events automatically triggers additional processing which aims at determining the relevance of the eliciting events [8, 50]. According to this assumption, distraction occurs as an automatic consequence of the perceptual processing of the auditory input. Therefore, distraction by auditory changes and auditory deviants should be primarily determined by perceptual characteristics of the auditory input (i.e., the degree to which it deviates from what is expected based on previous stimulation) and should be largely independent of the individual's emotional-motivational state [8, 21, 50]. This assumption leads to the prediction that both the auditory-deviant effect and the changing-state effect should be independent of positive and negative mood states.

### Ethics statement

The experiments were approved by the ethics committee of the Faculty of Mathematics and Natural Sciences at Heinrich Heine University Düsseldorf and were performed in accordance with the Declaration of Helsinki. All participants gave written informed consent before participating in the experiment.

### Experiment 1

In the first experiment, we tested the effect of positive mood on the auditory-deviant effect and the changing-state effect. To this end, happy and neutral mood states were induced using a

well-established mood-induction procedure consisting of a combination of autobiographical recall and music that is known to lead to powerful effects on the individuals' mood states [37, 51–53]. Within each mood group, a steady-state condition was contrasted with both an auditory-deviant condition and a changing-state condition in the standard serial-recall paradigm.

## Method

**Participants.** A total of 216 participants took part in the experiment in exchange for course credit or a monetary compensation of 4 €. The participants were recruited on campus at Heinrich Heine University Düsseldorf. The data sets of two participants had to be excluded prior to analysis because of technical errors. The final sample consisted of 214 participants (174 women). Using G*Power [54] we determined that, given a sample size of $N = 214$ and $\alpha = .05$, it was possible to detect an interaction between mood and distractor condition of the size $\eta_p^2 = .07$ with a statistical power of $1-\beta = .95$. All participants were fluent German speakers (189 native speakers) and reported normal hearing and normal or corrected-to-normal vision. Their age ranged from 18 to 40 years with a mean age of 23 years ($SD = 4$).

**Materials.** *Mood induction.* A combination of autobiographical recall and music was used to induce either a happy or a neutral mood. These mood-induction procedures are well established and known to be particularly effective [46–49]. The combination of both mood-induction procedures has been successfully used in previous studies on auditory distraction [37, 38]. Participants received the instructions for the autobiographical recall on a computer screen. The music was presented via headphones at about 60 dB (A) $L_{eq}$. In the happy mood condition, participants were asked to recall the happiest event of their life as vividly and with as much detail as possible for four minutes. Subsequently, they were asked to write down a detailed description of this event into a text field. The writing phase lasted five minutes. Participants listened to music during the whole autobiographical recall phase and throughout the subsequent mood assessment. The following musical pieces were played: *Eine kleine Nachtmusik* by Mozart, *Mazurka from Coppelia* by Delibes, and *Allegro from Brandenburg Concerto No. 2* by Bach. These musical pieces were selected for their capability to induce a happy mood [37, 51–53, 55]. The neutral-mood induction was identical to the happy-mood induction with the following exceptions. Participants recalled and wrote down details about their last visit to the grocery store. During the neutral-mood induction, participants listened to the following musical pieces: *The Planets–Neptune, the Mystic* by Gustav Holst and the *Largo movement from New World Symphony* by Antonin Dvorak. These musical pieces were selected for their capacity to induce a neutral mood [37, 38, 55–58].

*Mood assessment.* To verify that mood induction was effective, we used the German version of the Positive and Negative Affect Schedule (PANAS; [59, 60]). PANAS consists of 20 emotional adjectives, divided into two sub-scales, to measure positive affect (10 items) and negative affect (10 items) separately. Participants rated on a scale from 1 (very slightly) to 5 (extremely) to what extent the items reflected their current mood state. Additionally, participants rated their reaction to the musical pieces on the Self-Assessment Manakin (SAM; [61]). The SAM is a non-verbal assessment technique that consists of three five-point bipolar scales that serve to assess emotional reactions along the three dimensions valence (1 = unhappy to 5 = happy), arousal (1 = calm to 5 = excited), and dominance (1 = controlled to 5 = dominant).

*Serial-recall task.* A standard serial-recall task was used. In each trial, the visual to-be-remembered sequences consisted of eight digits randomly sampled from the set {1, 2, . . ., 9} without replacement. The digits were presented at a rate of 750 ms in black 80 pt Monaco font against white background at the centre of the screen of the computer that controlled the experiment. The auditory distractors consisted of a set of 12 one-syllable German words spoken by a

female voice. These words were recorded with a 44.1 sampling rate using 16-bit format. They were edited to last one second and normalized to minimize amplitude differences among the stimuli. Distractors were played at about 65 dB(A) $L_{eq}$. The same word set has been used in previous studies where robust auditory-deviant effects and changing-state effects have been observed [e.g., 62, 63]. For each steady-state sequence, one word was randomly drawn from the set and repeated eight times. Auditory-deviant sequences were identical to steady-state sequences except that the word at the fifth, sixth, or seventh position (selected randomly) was replaced by a different word from the set (the auditory-deviant). For changing-state trials, eight different words were randomly drawn from the word set without replacement. Simultaneously to the presentation of each target, a distractor word was presented.

**Procedure.** A 2 × 3 design was used with mood (neutral, happy) as group variable and distractor condition (steady state, auditory deviant, changing state) as repeated-measures variable. Serial-recall performance was used as the dependent variable. Based on the order of appearance in the lab, participants were alternately assigned to either the happy-mood group or the neutral-mood group. Following this procedure, half of the participants were assigned to either of the mood groups. First, participants performed 10 steady-state training trials to familiarize themselves with the serial-recall task. They were instructed to focus on the visually presented digits and to ignore the words presented over the headphones. They were told that the auditorily presented words were irrelevant for the task throughout the whole experiment. The data of the training trials were not analysed.

In the experiment proper (Fig 1), participants first completed the PANAS which served to measure their baseline mood (pre mood induction). Then, either a happy mood or a neutral mood was induced. After the mood induction, participants completed the PANAS a second time in order to measure mood changes due to the mood-induction procedure (post mood induction).Then, the serial-recall phase commenced. Participants completed eight steady-state trials, eight auditory-deviant trials and eight changing-state trials in a randomized order. This number of trials per condition is typical for experiments using the serial-recall paradigm [e.g., 8, 63, 64]. Each trial was initiated by pressing the space bar of the computer keyboard. After one second, the first to-be-remembered digit was shown. Throughout the presentation of the target sequence of digits, auditory distractors had to be ignored. Immediately after the presentation of the to-be-remembered target sequence, eight question marks appeared in the middle of the screen and had to be replaced by the remembered digits. The digits were consecutively typed using the number pad of the keyboard. Participants were not allowed to skip a digit or to correct a response. After all question marks were replaced, participants could continue with the next trial by pressing the space bar of the keyboard. The software running the experiment was written in LiveCode (Version 9, available at https://livecode.com). The whole experiment lasted about 30 minutes on average.

## Results

The data were analysed using the MANOVA approach to repeated-measures analyses [65]. In our application, all multivariate test criteria correspond to the same exact *F* statistic which is reported. Partial eta squared ($\eta_p^2$) is reported as an effect size measure. All analyses were carried out using IBM SPSS Statistics 27. The dataset of the experiment is available in the supplementary online material in the Open Science Framework repository at https://osf.io/tqjwr/.

**Mood assessment.** As a manipulation check, a rater who had been trained in data protection evaluated the answers participants provided during the autobiographical recall task to check whether they complied with the instructions. All participants in the neutral-mood group recalled a neutral situation of grocery shopping without any affective incidents. Participants in

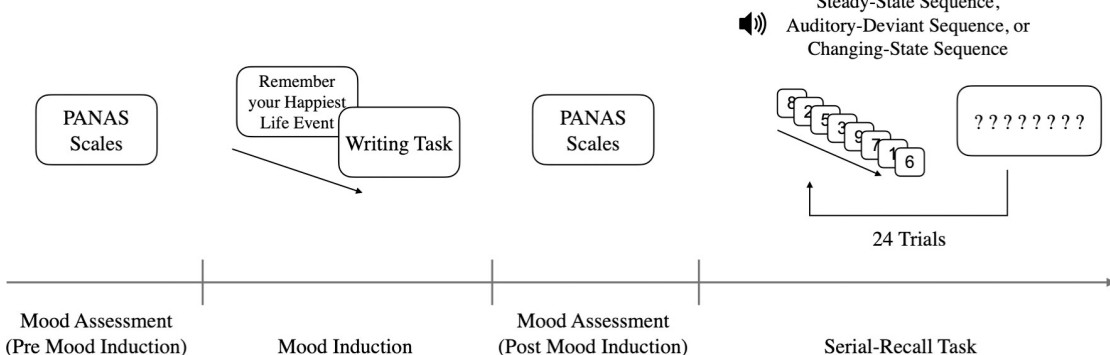

**Fig 1. Schematic illustration of the experimental procedure of Experiment 1 following the serial-recall training trials.** The happy mood-induction procedure is illustrated.

the happy-mood group all recalled positive events, mostly significant events with close friends, family or spouses, memories of vacations, or events associated with personal success such as academic or athletic achievements.

*PANAS positive-affect scores*. Overall, positive affect increased after happy-mood induction, $F(1, 212) = 69.28$, $p < .001$, $\eta_p^2 = .25$ (Fig 2A). The happy-mood group reported more positive affect than the neutral-mood group, $F(1, 212) = 34.97$, $p < .001$, $\eta_p^2 = .14$. These main effects were qualified by an interaction between time of testing and induced mood, $F(1, 212) = 41.38$, $p < .001$, $\eta_p^2 = .16$, suggesting that the increase in positive mood was more pronounced after happy-mood induction than after neutral-mood induction. Two supplementary analyses compared the positive-affect scores of the two mood-induction groups before and after the mood-induction procedure. The happy-mood group had a better mood than the neutral-mood group even before mood induction (probably because, other than in the following experiments, participants were instructed about the nature of the upcoming happy or neutral mood induction before these ratings were obtained), $F(1, 212) = 8.49$, $p = .004$, $\eta_p^2 = .04$, but there was a much stronger difference in positive mood between the neutral-mood group and the happy-mood group after the mood-induction procedure, $F(1, 212) = 56.97$, $p < .001$, $\eta_p^2 = .21$.

*PANAS negative-affect scores*. Happy-mood induction had no effect on the negative-affect scores of the PANAS. There was a main effect of time of testing, $F(1, 212) = 63.52$, $p < .001$, $\eta_p^2 = .23$, indicating that overall negative affect decreased during the mood-induction procedure. However, there was no effect of mood, $F(1, 212) = 2.71$, $p = .101$, $\eta_p^2 = .01$, and no interaction between time of testing and mood, $F(1, 212) = 0.20$, $p = .658$, $\eta_p^2 < .01$. Happy-mood induction thus had a selective effect on positive affect and did not significantly influence negative affect.

*SAM scores*. The happy music played during happy-mood induction was rated as more positive, $F(1, 212) = 14.20$, $p < .001$, $\eta_p^2 = .06$, and was associated with higher arousal, $F(1, 212) = 31.36$, $p < .001$, $\eta_p^2 = .13$, and higher dominance, $F(1, 212) = 32.87$, $p < .001$, $\eta_p^2 = .13$, than the neutral music that was played during neutral-mood induction (Table 1).

**Serial-recall performance.** As in previous studies [e.g., 50], a strict serial-recall criterion was used to measure serial-recall performance. In line with this criterion, only items that were recalled at the correct serial position were scored as correct. Serial-recall performance was affected by distractor condition, $F(2, 211) = 42.80$, $p < .001$, $\eta_p^2 = .29$. Mood had no main effect on serial-recall performance, $F(1, 212) < .01$, $p = .953$, $\eta_p^2 < .01$, and there was no interaction between distractor condition and mood, $F(2, 211) = 0.88$, $p = .417$, $\eta_p^2 = .01$ (Fig 3A). Two further analyses were conducted to analyse the auditory-deviant effect and the changing-

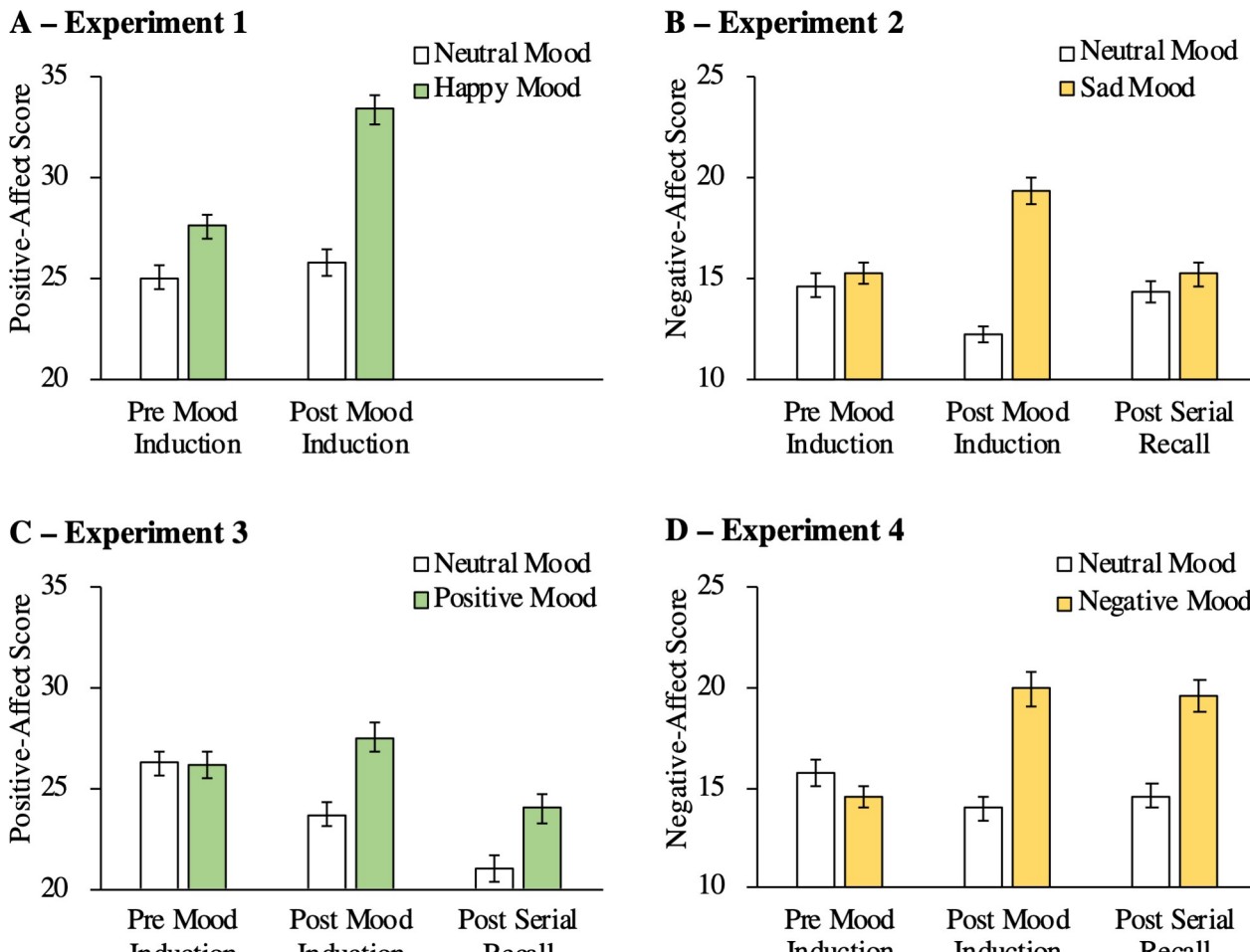

**Fig 2. Mean PANAS scores as a function of induced mood and time of testing.** PANAS scores can range from 10 to 50 points. The error bars represent the standard errors of the means. (A) Positive-affect scores of Experiment 1. (B) Negative-affect scores of Experiment 2. (C) Positive-affect scores of Experiment 3. (D) Negative-affect scores of Experiment 4.

state effect separately from each other. When contrasting the steady-state condition with the auditory-deviant condition, there was evidence of an auditory-deviant effect, $F(1, 212) = 19.86$, $p < .001$, $\eta_p^2 = .09$, but mood had no influence on the size of the auditory-deviant effect, $F(1, 212) = 1.77$, $p = .185$, $\eta_p^2 = .01$. When contrasting the steady-state condition with the changing-state condition, there was evidence of a changing-state effect, $F(1, 212) = 85.48$, $p < .001$, $\eta_p^2 = .29$, but the changing-state effect was not affected by mood, $F(1, 212) = 0.54$, $p = .465$, $\eta_p^2 < .01$.

## Discussion

The mood-induction procedure used in Experiment 1 was successful in inducing a positive mood in the happy-mood group while the control group remained in a neutral mood state. There was a mean difference of 7.64—associated with an effect size of $\eta_p^2 = .21$—for the PANAS positive-affect score between the happy mood-induction group and the neutral mood-induction group post mood induction. The difference of PANAS positive-affect scores between participants in the happy-mood group and those of the control group was as large as the corresponding difference obtained in other mood-induction studies (e.g., a difference of

**Table 1. Means and standard deviations of the SAM ratings of the mood-inducing music (Experiment 1, 2) and pictures (Experiment 3, 4).**

| Group | Valence | | Arousal | | Dominance | |
|---|---|---|---|---|---|---|
| | M | SD | M | SD | M | SD |
| Experiment 1 | | | | | | |
| Neutral Mood | 3.58 | 0.90 | 1.96 | 0.98 | 3.04 | 0.62 |
| Happy Mood | 4.03 | 0.81 | 2.77 | 1.12 | 3.56 | 0.70 |
| Experiment 2 | | | | | | |
| Neutral Mood | 3.68 | 0.81 | 1.75 | 0.87 | 3.04 | 0.79 |
| Sad Mood | 2.12 | 0.77 | 2.45 | 0.99 | 2.54 | 0.79 |
| Experiment 3 | | | | | | |
| Neutral Mood | 3.09 | 0.27 | 2.36 | 0.57 | – | – |
| Positive Mood | 4.20 | 0.39 | 2.34 | 0.50 | – | – |
| Experiment 4 | | | | | | |
| Neutral Mood | 3.12 | 0.34 | 2.17 | 0.64 | – | – |
| Negative Mood | 1.53 | 0.39 | 4.01 | 0.60 | – | – |

The five-point bipolar scales of SAM were used. SAM scores can range from unhappy (1) to happy (5) for valence, from calm (1) to excited (5) for arousal, and from controlled (1) to dominant (5) for dominance.

5.68 on the PANAS positive-affect score in the study of Pacheco-Unguetti and Parmentier [37]), supporting the evidence that the combination of autobiographical recall and mood-congruent music is powerful to induce positive mood states [e.g., 37, 48]. What is more, the PANAS positive-affect sum score was increased by 5.8 points after the happy-mood induction (Fig 2A) which corresponds to a difference of 0.58 on the PANAS positive-affect mean score. This difference is in line with results of a meta-analysis identifying an average increase pre versus post positive-mood induction of 0.29 of the mean scores on the PANAS positive-affect scale [47]. Based on these results, it can be concluded that the induction of a positive mood was comparatively effective.

Nevertheless, distraction was not modulated by mood state. While Experiment 1 successfully replicated the auditory-deviant effect [12, 13] as well as the changing-state effect [15, 16], neither the size of the auditory-deviant effect nor the size of the changing-state effect was larger when participants were in a happy mood compared to when they were in a neutral mood. The current experiment thus indicates that auditory distraction in a serial-recall task is not enhanced or otherwise affected by a happy mood compared to a neutral mood. However, it remains yet to be tested whether negative mood influences auditory distraction. For this purpose, we replaced the happy-mood induction by a sad-mood induction in Experiment 2 using the same procedure as in Experiment 1. As an improvement of the procedure, we added a mood assessment following the serial-recall task to test whether the induced mood persisted until the end of the experiment.

## Experiment 2

### Method

**Participants.** Prior to analysis, nine data sets had to be removed because nine participants had participated twice. Only participants who had not participated in the previous experiment (Experiment 1) were allowed to participate. The remaining sample consisted of 210 participants (143 women) recruited on campus at Heinrich Heine University Düsseldorf. Given a sample size of $N = 210$ and $\alpha = .05$, it was possible to detect an interaction between mood and distractor condition of the size $\eta_p^2 = .07$ with a statistical power of $1-\beta = .95$. The participants

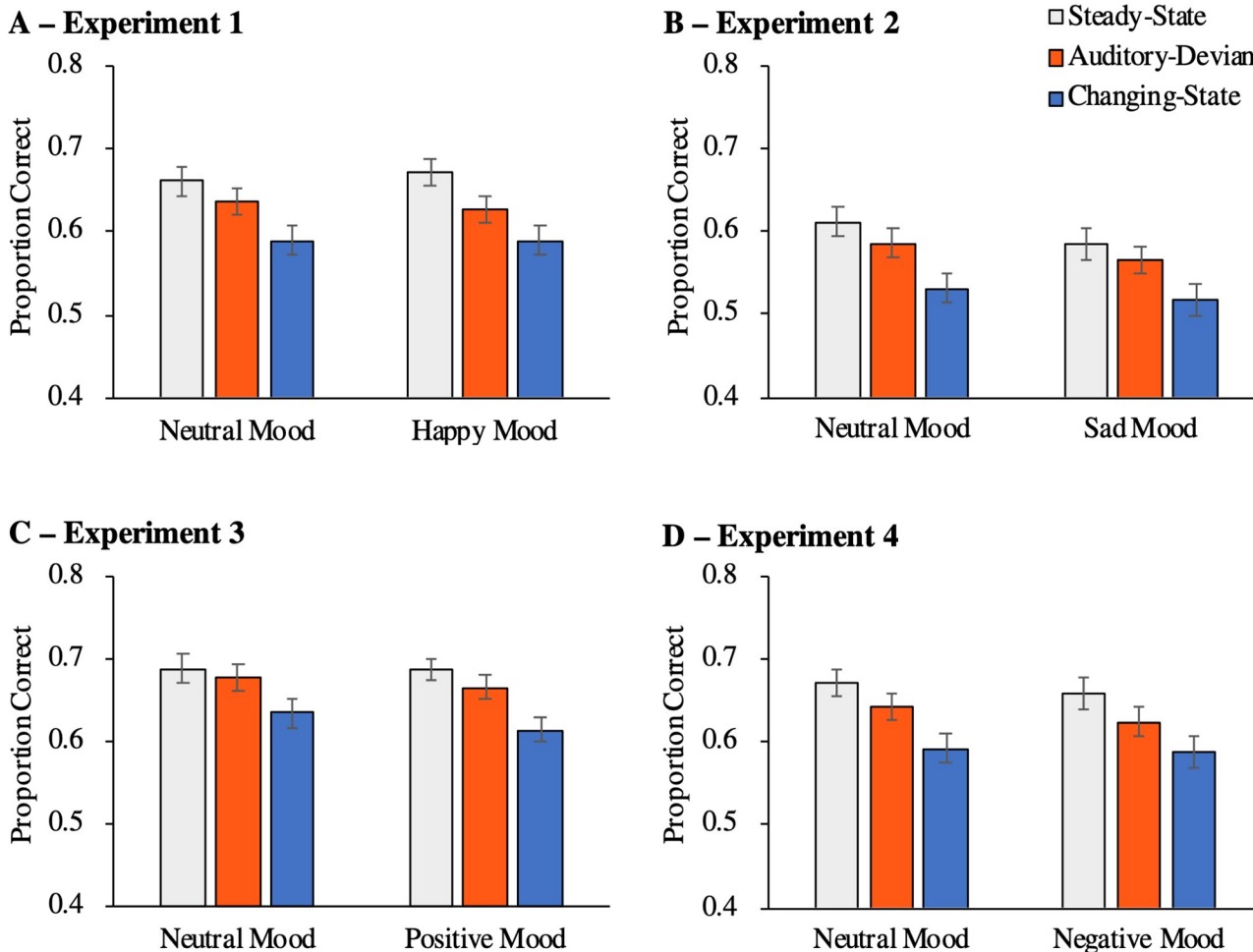

**Fig 3. Proportion of correct responses in serial recall as a function of distractor condition and mood.** The error bars represent the standard errors of the means. (A) Results of Experiment 1. (B) Results of Experiment 2. (C) Results of Experiment 3. (D) Results of Experiment 4.

received a small monetary compensation of 4 € or course credit for participation. All participants were fluent German speakers and reported normal hearing and normal or corrected-to-normal vision. Their age ranged from 17 to 42 years with a mean age of 23 years ($SD = 4$).

**Materials and procedure.** Materials and procedure were identical to those used in Experiment 1, with the following exceptions. The happy-mood induction was replaced by a sad-mood induction. A $2 \times 3$ design was used with mood (neutral, sad) as group variable and distractor condition (steady state, auditory deviant, changing state) as repeated-measures variable. To induce a sad mood, participants were asked to remember and write down the saddest event of their life while they listened to mood-congruent sad music. The musical pieces, *Adagio for Strings*, *Op. 11* by Samuel Barber and *5th Symphony Adagietto* by Mahler, are evidently capable to induce sadness [38, 51, 53, 55, 58]. We also added a final mood assessment in which participants rated their negative and positive affect using the German version of PANAS [59, 60] after the final trial of the serial-recall task. At the end of the experiment, all participants listened to *Eine kleine Nachtmusik* by Mozart which served to help participants to return to a positive mood before they were debriefed and dismissed (see also [38]).

## Results

**Mood assessment.** As a manipulation check, a rater who was trained in data protection evaluated the answers participants provided during the autobiographical recall task to check whether they complied with the instructions. All participants in the neutral-mood group reported a neutral situation of grocery shopping without any emotional events. Participants in the sad-mood group all recalled negative events, such as the loss of family members or close friends, the end of a romantic relationship, family conflicts, health issues, or situations associated with academic failure.

*PANAS negative-affect scores.* Negative affect differed as a function of time of testing, $F(2, 207) = 4.90$, $p = .008$, $\eta_p^2 = .05$ (Fig 2B). Overall, the sad-mood group reported more negative affect than the neutral-mood group, $F(1, 208) = 17.31$, $p < .001$, $\eta_p^2 = .08$. These main effects were qualified by an interaction between time of testing and induced mood, $F(2, 207) = 51.14$, $p < .001$, $\eta_p^2 = .33$. Negative affect did not differ between groups before mood induction, $F(1, 208) = 0.64$, $p = .426$, $\eta_p^2 < .01$. After mood induction, the sad-mood group reported more negative affect than the neutral-mood group, $F(1, 208) = 78.88$, $p < .001$, $\eta_p^2 = .27$, but there was only a non-significant trend towards more negative affect in the sad-mood group in comparison to the neutral-mood group at the end of the experiment, $F(1, 208) = 1.22$, $p = .271$, $\eta_p^2 = .01$.

*PANAS positive-affect scores.* Positive affect differed as a function of time of testing, $F(2, 207) = 73.53$, $p < .001$, $\eta_p^2 = .42$. Overall, the sad-mood group reported less positive affect than the neutral-mood group, $F(1, 208) = 14.23$, $p < .001$, $\eta_p^2 = .06$. These main effects were qualified by an interaction between time of testing and induced mood, $F(2, 207) = 31.92$, $p < .001$, $\eta_p^2 = .24$, suggesting that the decrease of positive mood after mood induction was more pronounced for the sad-mood group than for the neutral-mood group. Positive affect did not differ between groups before mood induction, $F(1, 208) = 1.42$, $p = .235$, $\eta_p^2 = .01$. After mood induction, positive affect was lower after sad-mood induction than after neutral-mood induction, $F(1, 208) = 50.80$, $p < .001$, $\eta_p^2 = .20$. At the end of the experiment, there was only a non-significant trend towards positive affect being lower in the sad-mood group in comparison to the neutral-mood group, $F(1, 208) = 3.02$, $p = .084$, $\eta_p^2 = .01$.

*SAM scores.* The sad music that was played during sad-mood induction was rated as more negative, $F(1, 208) = 202.17$, $p < .001$, $\eta_p^2 = .49$, and was associated with higher arousal, $F(1, 208) = 29.98$, $p < .001$, $\eta_p^2 = .13$, and lower dominance ratings than the neutral music that was played during neutral-mood induction, $F(1, 208) = 20.99$, $p < .001$, $\eta_p^2 = .09$ (Table 1).

**Serial-recall performance.** Serial-recall performance differed as a function of distractor condition, $F(2, 207) = 39.85$, $p < .001$, $\eta_p^2 = .28$. Mood had no main effect on performance, $F(1, 208) = 0.83$, $p = .363$, $\eta_p^2 < .01$, and there was no interaction between distractor condition and mood, $F(2, 207) = 0.25$, $p = .775$, $\eta_p^2 < .01$ (Fig 3B). Two further analyses were conducted to analyse the auditory-deviant effect and the changing-state effect separately. When contrasting the steady-state condition with the auditory-deviant condition, there was evidence of an auditory-deviant effect, $F(1, 208) = 7.24$, $p = .008$, $\eta_p^2 = .03$, but mood had no influence on the size of the auditory-deviant effect, $F(1, 208) = 0.14$, $p = .712$, $\eta_p^2 < .01$. Contrasting the steady-state condition with the changing-state condition, there was evidence of a changing-state effect, $F(1, 208) = 71.07$, $p < .001$, $\eta_p^2 = .25$, but the size of the changing-state effect was not modulated by mood, $F(1, 208) = 0.51$, $p = .478$, $\eta_p^2 < .01$.

## Discussion

In Experiment 2, the mood-induction procedure was successful to induce sad mood in the sad-mood group while the other group remained in a neutral mood state. There was a mean

difference of 7.08—with an effect size of $\eta_p^2 = .27$—on the PANAS negative-affect score between the sad mood-induction group and the neutral mood-induction group immediately after mood induction. These results are in line with the mood differences measured in other studies in which negative mood was induced with a combination of autobiographical recall and music (e.g., a difference of 6.75 on the PANAS negative-affect score in the study of Pacheco-Unguetti and Parmentier [38]). Furthermore, there was a mean difference of 4.1 for the sum score of the PANAS negative-affect scale before and after the sad-mood induction (Fig 2B) which equals a difference of 0.41 of the mean PANAS negative-affect score. This difference is in line with results of a meta-analysis identifying a typical difference pre and post negative-mood induction of 0.45 of the mean scores of PANAS negative-affect scale [47]. The effectiveness of the present procedure to induce negative moods is thus in line with previous studies. Nevertheless, there was again no evidence of a modulation of distraction by mood in Experiment 2. While we successfully replicated the auditory-deviant effect as well as the changing-state effect, neither the size of the auditory-deviant effect nor the size of the changing-state effect differed in sad mood compared to neutral mood.

Experimentally induced mood states are known to be rather volatile [e.g., 48, 66]. As is typical for mood-induction procedures [e.g., 67, 68], strong effects on mood were observed directly after the mood induction, but the mood states did not persist until the end of the experiment. Therefore, we needed to develop a mood-induction procedure that enabled us to continuously control and monitor mood state. To this end, we showed participants pictures of positive, negative, and neutral scenes, and asked them to imagine themselves in the depicted scenes and to rate how happy or unhappy and how calm or excited they would feel. Pictorial stimuli are known to be successful in inducing positive as well as negative mood states [e.g., 47, 69]. The main advantage of this procedure is that it is easily possible to repeat the mood-induction procedure immediately before each serial-recall trial to maintain participants in a positive or negative mood until the end of the serial-recall task. Parallel to Experiment 1, we started by comparing positive mood to neutral mood in Experiment 3.

## Experiment 3

### Method

**Participants.**   Two hundred and thirty participants (162 women) recruited on campus at Heinrich Heine University Düsseldorf participated in exchange for a monetary compensation of 4 € or course credit. Given a sample size of $N = 230$ and $\alpha = .05$, it was possible to detect an interaction between mood and distractor condition of the size $\eta_p^2 = .06$ with a statistical power of $1-\beta = .95$. As in the previous experiments, all participants were fluent German speakers and reported normal hearing as well as normal or corrected-to-normal vision. Their age ranged from 18 to 39 years with a mean age of 22 ($SD = 4$) years.

**Materials.**   The mood assessment and the serial-recall task were the same as in the previous experiments. However, a different mood-induction procedure was used to ensure that participants stayed in a neutral or positive mood until the end of the experiment. To this end, a set of 34 neutral pictures and a set of 34 positive pictures were chosen from the International Affective Picture System (IAPS; [70]). Neutral pictures were selected to be of neutral valence and low arousal. The selected set of neutral pictures showed people with neutral facial expressions and household articles, among other neutral objects. Positive pictures were selected to have a maximally positive valence rating. Pictures showing sexual content were excluded as people may show ambivalent emotional responses to erotic stimuli, at least when encountering them in a lab environment. The positive pictures showed, for example, smiling babies, landscapes, or happy families. The valence and arousal ratings of the positive pictures were

**Table 2. Means and standard deviations of the normative ratings of the picture sets taken from the IAPS [70].**

| Picture Set | Valence | | Arousal | |
|---|---|---|---|---|
| | *M* | *SD* | *M* | *SD* |
| Positive Picture Set | 7.93 | 0.20 | 4.84 | 0.80 |
| Neutral Picture Set | 4.99 | 0.14 | 2.63 | 0.35 |
| Negative Picture Set | 2.15 | 0.23 | 6.05 | 0.83 |

The normative ratings were assessed using the nine-point bipolar scales for valence and arousal of SAM, ranging from 1 (lowest rating) to 9 (highest rating). Hence, higher scores represent a higher rating on each dimension (higher valence, higher arousal).

significantly higher than the ratings for the neutral pictures, $F(1, 66) = 4862.19$, $p < .001$, $\eta_p^2 = .99$ and $F(1, 66) = 217.64$, $p < .001$, $\eta_p^2 = .77$, respectively (Table 2). The list of the positive and neutral pictures that were used in the present experiment is available in the Open Science Framework repository at https://osf.io/tqjwr/.

**Procedure.** As in the previous experiments, a 2 × 3 design was used with mood (neutral, positive) as group variable and distractor condition (steady state, auditory deviant, changing state) as repeated-measures variable. Again, participants started with 10 steady-state training trials to familiarize themselves with the serial-recall task.

The experiment proper (Fig 4) started with the completion of the first PANAS [59, 60] to measure baseline mood (pre mood induction). Immediately after the PANAS had been completed, the massed-mood-induction procedure started. Depending on mood condition, participants saw 10 positive or 10 neutral pictures taken from the IAPS database. The pictures were presented at a size of 1024 × 768 pixels at the centre of the computer screen. The participants were instructed to put themselves into the presented situations and imagine the feelings and thoughts they would experience in the depicted situations. After five seconds, the SAM valence scale [61] was shown directly below the picture. As soon as participants had rated the valence of their feelings in the depicted situation, the valence scale was replaced by the arousal scale. Participants had a total of 10 seconds to rate how happy or unhappy and how calm or excited they would be in the depicted situation. Each picture stayed on screen for all of the 15 seconds. Then the picture and the SAM scale were replaced by a fixation cross. After one second, the next picture was presented automatically. Following the massed-mood-induction procedure, participants completed the PANAS a second time (post mood induction). Next, the serial-recall phase started. The procedure of the serial-recall task was the same as in the previous

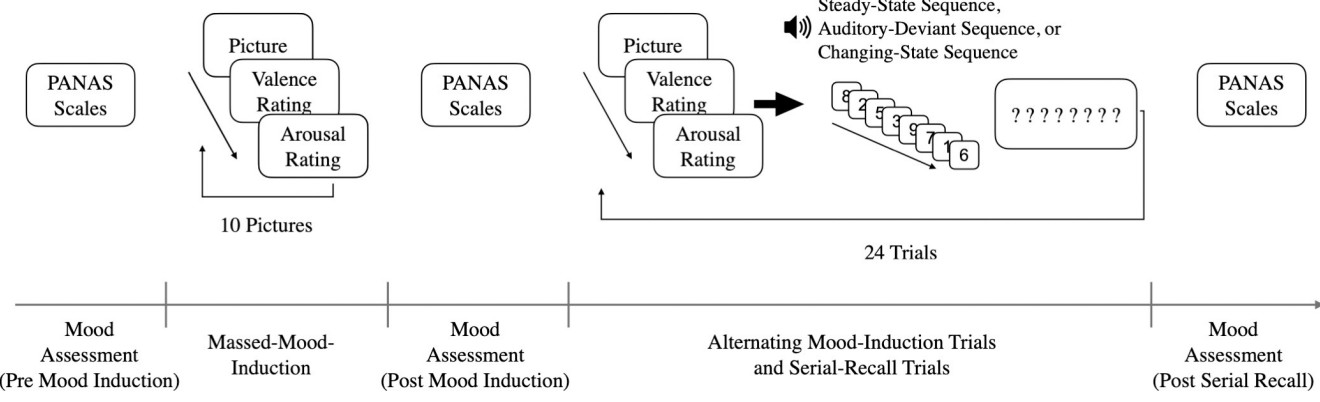

**Fig 4. Schematic illustration of the experimental procedure of Experiment 3 following the serial-recall training trials.**

experiments. However, each serial-recall trial was preceded by a further mood-induction trial in which participants saw, depending on the mood condition, a neutral or positive picture, imagined themselves being in the neutral or positive situation that was depicted, and rated how happy or unhappy, and how calm or exited they would feel in this situation. Over the whole experiment, pictures were randomly drawn without replacement from the respective picture set. Thus, depending on the mood condition, the participants saw either all neutral pictures or all positive pictures once. After the last serial-recall trial, participants answered PANAS a third time (post serial recall). The whole experiment lasted about 30 minutes on average.

## Results

**Mood assessment.** *PANAS positive-affect scores.* Positive affect differed as a function of time of testing, $F(2, 227) = 69.95$, $p < .001$, $\eta_p^2 = .38$ (Fig 2C). Overall, the positive-mood group reported more positive affect than the neutral-mood group, $F(1, 228) = 6.08$, $p = .014$, $\eta_p^2 = .03$. These main effects were qualified by an interaction between time of testing and induced mood, $F(2, 227) = 20.96$, $p < .001$, $\eta_p^2 = .16$. Positive-affect scores did not differ between groups before mood induction, $F(1, 228) = 0.01$, $p = .918$, $\eta_p^2 < .01$. After mood induction, the positive-mood group had more positive affect than the neutral-mood group, $F(1, 228) = 15.19$, $p < .001$, $\eta_p^2 = .06$. At the end of the experiment, the positive-mood group still reported more positive affect than the neutral-mood group, $F(1, 228) = 8.03$, $p = .005$, $\eta_p^2 = .03$, suggesting that differences in positive affect between the positive-mood group and the neutral-mood group were still present at the end of the experiment.

*PANAS negative-affect scores.* Negative affect differed as a function of time of testing, $F(2, 227) = 28.93$, $p < .001$, $\eta_p^2 = .20$ indicating a slight decrease of overall negative affect over the course of the experiment. However, there was no main effect of mood group, $F(1, 228) = 1.30$, $p = .255$, $\eta_p^2 = .01$, and no interaction between time of testing and induced mood, $F(2, 227) = 1.51$, $p = .224$, $\eta_p^2 = .01$. The positive-mood induction thus had a selective effect on positive affect and did not influence negative affect.

*SAM scores.* The overall rate of missing responses to the SAM scales during mood induction was low, with a mean of 1.6% missing answers, indicating that 10 seconds were enough time for the ratings. About half of all missing responses occurred in the first two ratings of the initial mood-induction block, indicating that participants had to get familiar with the timed rating task but adapted quickly. Participants indicated that they would have happier feelings in the positive scenes than in the neutral scenes, $F(1, 228) = 633.08$, $p < .001$, $\eta_p^2 = .74$, while arousal ratings did not differ as a function of the type of pictures, $F(1, 228) = 0.11$, $p = .742$, $\eta_p^2 < .01$ (Table 1).

**Serial-recall performance.** Serial-recall performance differed as a function of distractor condition, $F(2, 227) = 36.91$, $p < .001$, $\eta_p^2 = .25$. Mood had no main effect on performance, $F(1, 228) = 0.27$, $p = .603$, $\eta_p^2 < .01$, and there was no interaction between distractor condition and mood, $F(2, 227) = 0.72$, $p = .486$, $\eta_p^2 = .01$ (Fig 3C). When the steady-state condition was contrasted with the auditory-deviant condition, there was evidence of an auditory-deviant effect, $F(1, 228) = 5.41$, $p = .021$, $\eta_p^2 = .02$, but mood had no influence on the size of the auditory-deviant effect, $F(1, 228) = 0.39$, $p = .532$, $\eta_p^2 < .01$. When the steady-state condition was contrasted with the changing-state condition, there was evidence of a changing-state effect, $F(1, 228) = 67.82$, $p < .001$, $\eta_p^2 = .23$, but the size of the changing-state effect did not differ as a function of mood, $F(1, 228) = 1.45$, $p = .229$, $\eta_p^2 = .01$.

## Discussion

The mood-induction procedure was successful in the sense that it caused participants in the positive-mood group to be in a more positive mood than the group that saw neutral pictures. Descriptively, the effect was somewhat less pronounced than the positive-mood induction used in Experiment 1, but the difference in the positivity of mood-ratings of the two respective groups was still present at the end of the experiment. Even though the auditory-deviant effect and the changing-state effect were successfully replicated, neither of the two effects was modulated by mood state.

In Experiment 3, positive mood decreased over the course of the experiment so that participants were in a more positive mood before starting the demanding serial-recall task than at the end of the experiment. We therefore ran an additional analysis in which we included time course (operationalized as ordinal trial number) as a factor to check whether susceptibility to distraction changed over the course of the experiment. However, this analysis revealed that there was no main effect of time course on serial recall, $F(7, 222) = 1.30$, $p = .250$, $\eta_p^2 = .04$, no two-way interaction between distraction and time course, $F(14, 215) = 1.35$, $p = .182$, $\eta_p^2 = .08$, and no three-way interaction between mood, distraction, and time course, $F(14, 215) = 1.04$, $p = .418$, $\eta_p^2 = .06$. These results thus indicate that changes in emotional state over the course of the experiment did not affect baseline performance or susceptibility to distraction.

In Experiment 4, negative mood was contrasted with neutral mood as in Experiment 2. However, we now relied on the same mood-induction procedure as in Experiment 3 to maintain differences in emotional mood states until the end of the serial-recall task. Another notable difference is that participants in Experiment 2 were induced into sad mood, which is a negative mood state with low arousal [71] but participants of Experiment 4 were confronted with threatening pictorial scenes, which are known to have negative valence and high arousal [70]. This allowed us to test whether negative stimuli with high arousal may cause a narrowing of attention [72, 73] and thus decrease auditory distraction.

## Experiment 4

### Method

**Participants.**   Only participants who had not participated in the previous experiment (Experiment 3) were allowed to participate. One hundred ninety-seven participants (146 women) recruited on campus at Heinrich Heine University participated in the experiment. Given a sample size of $N = 197$ and $\alpha = .05$, it was possible to detect an interaction between mood and distractor condition of the size $\eta_p^2 = .07$ with a statistical power of $1-\beta = .95$. Participants received a small monetary payment of 4 € or course credit in exchange for participation. All participants were fluent German speakers and reported normal hearing as well as normal or corrected-to-normal vision. Their age ranged from 18 to 55 years with a mean age of 23 ($SD = 4$) years.

**Materials and procedure.**   Materials and procedure were the same as those of Experiment 3 with the following exceptions. A $2 \times 3$ design was used with mood (neutral, negative) as group variable and distractor condition (steady state, auditory deviant, changing state) as repeated-measures variable. To induce negative mood, 34 negative pictures were drawn from the IAPS [70] showing unpleasant and threatening scenes like accidents, attacks, diseases, and injuries. The pictures were chosen to have a negative valence and high arousal. The same neutral pictures were used as in Experiment 3. The negative picture set was associated with significantly lower valence, $F(1, 66) = 3763.93$, $p < .001$, $\eta_p^2 = .98$, and significantly higher arousal, $F(1, 66) = 485.24$, $p < .001$, $\eta_p^2 = .88$, than the neutral pictures (Table 2). The list of the negative

and neutral pictures used in the present experiment is available in the supplementary online material at https://osf.io/tqjwr/. At the end of the experiment, all participants listened to *Eine kleine Nachtmusik* by Mozart to diminish the effects of the negative-mood induction before they were debriefed and dismissed.

## Results

**Mood assessment.** *PANAS negative-affect scores*. Negative affect differed as a function of time of testing, $F(2, 194) = 15.94$, $p < .001$, $\eta_p^2 = .14$ (Fig 2D). Overall, the negative-mood group reported more negative affect than the neutral-mood group, $F(1, 195) = 13.87$, $p < .001$, $\eta_p^2 = .07$. These main effects were qualified by an interaction between time of testing and mood, $F(2, 194) = 51.89$, $p < .001$, $\eta_p^2 = .35$. Negative affect did not differ between groups before mood induction, $F(1, 195) = 1.90$, $p = .169$, $\eta_p^2 = .01$. After mood induction, negative affect was higher in the negative-mood group than in the neutral-mood group, $F(1, 195) = 33.15$, $p < .001$, $\eta_p^2 = .15$, indicating that mood induction was successful. At the end of the experiment, negative affect was still higher in the negative-mood group than in the neutral-mood group, $F(1, 195) = 24.80$, p $< .001$, $\eta_p^2 = .11$, suggesting that we succeeded in keeping participants in the negative-mood group in a negative mood until the end of the serial-recall task.

*PANAS positive-affect scores*. Positive affect differed as a function of time of testing, $F(2, 194) = 85.29$, $p < .001$, $\eta_p^2 = .47$. There was no main effect of induced mood, $F(1, 195) = 0.10$, p $= .747$, $\eta_p^2 < .01$. However, there was an interaction between time of testing and mood, $F(2, 194) = 3.10$, $p = .047$, $\eta_p^2 = .03$, possibly due to the fact that the decrease in positive mood was slightly more pronounced in the negative-mood group than in the neutral-mood group. However, supplementary analyses comparing the positive affect scores of the two mood-induction groups separately before and after mood induction as well as at the end of the experiment showed no significant differences between the positive-affect scores of the two groups at all times of testing (all *p*'s $> .05$). The results thus suggest that the negative-mood induction primarily affected negative mood.

*SAM scores*. As in Experiment 3, the overall rate of missing responses to the SAM scales during mood induction was low with a mean of 1.8% missing answers. Participants indicated that they imagined feeling more unhappy in the negative scenes than in the neutral scenes, $F(1, 195) = 957.93$, $p < .001$, $\eta_p^2 = .83$, and that they felt higher arousal when imagining themselves in the negative scenes than when imagining themselves in the neutral scenes, $F(1, 195) = 433.33$, $p < .001$, $\eta_p^2 = .69$ (Table 1).

**Serial-recall performance.** Serial-recall performance differed as a function of distractor condition, $F(2, 194) = 41.78$, $p < .001$, $\eta_p^2 = .30$. Mood had no influence on task performance, $F(1, 195) = 0.28$, $p = .597$, $\eta_p^2 < .01$. There was no interaction between distractor condition and mood, $F(2, 194) = 0.45$, $p = .637$, $\eta_p^2 < .01$ (Fig 3D). When the steady-state condition was contrasted with the auditory-deviant condition, there was evidence of an auditory-deviant effect, $F(1, 195) = 17.66$, $p < .001$, $\eta_p^2 = .08$, but the auditory-deviant effect did not differ as a function of mood, $F(1, 195) = 0.05$, $p = .819$, $\eta_p^2 < .01$. When the steady-state condition was contrasted with the changing-state condition, there was evidence of a changing-state effect, $F(1, 195) = 83.23$, $p < .001$, $\eta_p^2 = .30$, but the changing-state effect did not differ as a function of mood either, $F(1, 195) = 0.43$, $p = .515$, $\eta_p^2 < .01$.

## Discussion

The negative-mood induction was successful as it caused participants in the negative-mood group to be in a more negative emotional state than the participants who saw neutral pictures.

It seems noticeable that this effect was about as pronounced as the effect of the well-established negative-mood induction procedure used in Experiment 2. However, in contrast to Experiment 2, negative mood persisted until after the serial-recall task. Hence, it can be concluded that the picture-based mood induction was powerful enough to induce and maintain participants in a negative mood. What is more, the negative-mood induction affected not only valence but also arousal. However, despite the different levels of arousal and the fact that the negative mood lasted until the end of the experiment, there was no evidence that distraction differed between participants in negative and those in neutral mood. Neither the size of the auditory-deviant effect nor the size of the changing-state effect was influenced by mood state.

## General discussion

The present series of experiments served to test whether emotional states are a major determinant of auditory distraction. It has been suggested that auditory distraction reflects a delicate trade-off between openness and selectivity [e.g., 5–7]. A priori, it seemed possible that the balance of these conflicting goals might be determined by mood state. Specifically, it has been suggested that positive mood leads to a broadening of the attentional focus and a loosening of the inhibitory control over distracting information [e.g., 25–27, 35] while negative mood leads to a narrowing of the attentional focus and an increase in inhibitory control of distracting information [e.g., 30, 31, 36]. From this theoretical framework, we derived the predictions that auditory distraction should increase in positive mood states and decrease in negative mood states but we noted that the available literature regarding cross-modal auditory distraction currently does not clearly support these predictions [37–39]. In the present study, we extend the range of available tests by examining whether two benchmark findings of working memory [17]—the auditory-deviant effect [12, 13] and the changing-state effect [15, 16]—are differentially affected by negative, positive, and neutral mood states. Contrasting the effects of mood on the auditory-deviant effect with those on the changing-state effect is interesting because the duplex-mechanism account of auditory distraction predicts that only the auditory-deviant effect should be affected by the emotional mood state of the individual while the changing-state effect should occur as an automatic consequence of the obligatory processing of the auditory input and thus should remain unaffected by emotional mood state [22]. However, the present results are in contrast to these predictions. Even though auditory distraction was robustly observed in all of the four experiments, neither the auditory-deviant effect nor the changing-state effect was affected by the manipulations of emotional mood states. The present results thus do not confirm that attentional diversion by auditory distractors—reflected by the size of the auditory-deviant effect—is influenced by positive or negative mood. This holds true even for emotional mood states with enhanced arousal (Experiment 4). The results thus support the conceptualization of auditory distraction as a primarily stimulus-driven process that is prevalent in related fields of research [e.g., 3, 4]. Specifically, the results are in line with an automatic-capture account according to which both the auditory-deviant effect and the changing-state effect arise in an automatic fashion from the obligatory perceptual processing of changes and deviations in the to-be ignored auditory channel [8, 21, 50].

To examine the possible influence of mood states on auditory distraction, it was necessary to induce a positive or a negative mood. Therefore, it is crucial that we were able to successfully induce positive (Experiment 1 and 3) and negative mood states (Experiment 2 and 4) in the present experiments. The size of the effects of the mood-induction procedures on reported mood were the order of magnitude of those reported in a recent meta-analysis [47]. In Experiments 1 and 2, we used the same mood-induction procedure as Pacheco-Unguetti and Parmentier [37, 38], but, contrary to their results, the induced mood did not persist until the end

of the experiment (Experiment 2). Therefore, we switched to a different mood-induction procedure in Experiments 3 and 4. This procedure had the advantage that the mood induction (imagining oneself in emotional scenes) could be repeated immediately before each trial of the serial-recall task. The procedure was effective in producing differences in mood that lasted until after the serial-recall phases of Experiments 3 and 4. Nevertheless, all of the experiments consistently showed that auditory distraction was independent of mood state.

Recall that in Experiment 3 we additionally analysed whether the susceptibility to distraction changed over the course of the experiment because the intensity of the positive mood decreased over the course of time in both mood groups. Although the results did not indicate that changes in emotional state over the course of the experiment affected baseline performance or susceptibility to distraction, it is also interesting to check for a possible variation in serial-recall performance and distraction as a function of the time course and induced mood in Experiments 1, 2 and 4. This check is particularly interesting for Experiments 1 and 2, as we did not assess the mood after the serial-recall task (Experiment 1) or the induced mood did not last until after the serial-recall task (Experiment 2). Hence, it may be possible that the induced mood affected auditory distraction at the beginning of the serial-recall task but the influence vanished as the induced mood declined over the course of time leading to a non-significant effect of mood on auditory distraction in the main analyses. Therefore, we also ran additional analyses in which we included time course (operationalized as ordinal trial number) as a factor for Experiments 1, 2 and 4. However, in line with the results of Experiment 3, time course did not interact with any of the other variables in these three experiments.

To explore whether mood has an effect on distraction when using an even larger sample size to achieve a higher statistical sensitivity while maintaining statistical power at a high level, we combined the serial-recall data of all four experiments (Fig 5). Given a total sample size of $N = 851$ and $\alpha = .05$, it was possible to detect an interaction between mood (positive, negative, neutral) and distractor condition as small as about $\eta_p^2 = .02$ with a statistical power of $1-\beta = .95$ in the combined analysis. As in the four individual experiments, serial-recall performance was affected by distractor condition, $F(2, 847) = 143.40$, $p < .001$, $\eta_p^2 = .25$. Mood had an

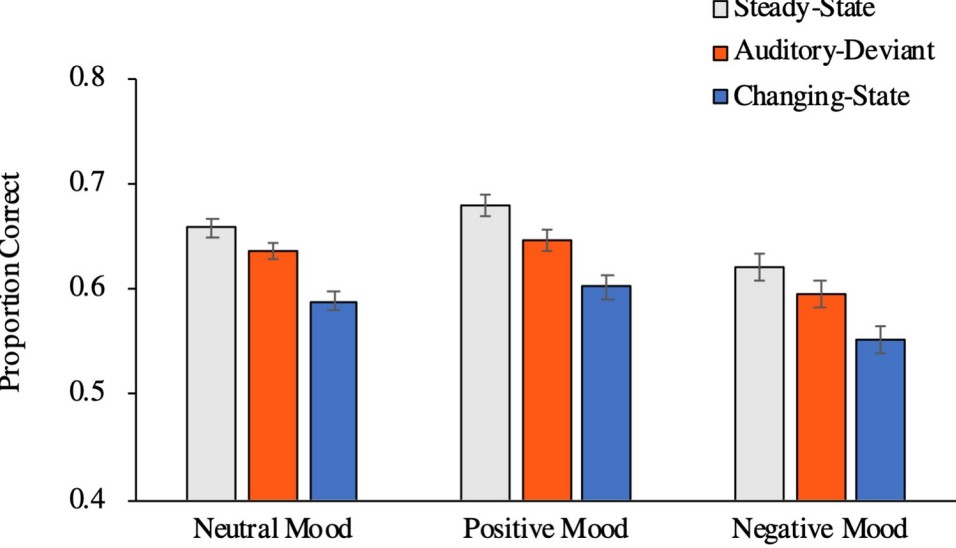

**Fig 5. Proportion of correct responses in serial recall for the combined data of all four experiments.** The proportion of correct responses is presented as a function of distractor condition and valence of induced mood. The error bars represent the standard errors of the means.

influence on serial-recall performance, $F(2, 848) = 6.33$, $p = .002$, $\eta_p^2 = .01$, which is due to the fact that serial-recall performance was somewhat lower in Experiment 2 than in the other experiments (Fig 3). More importantly, however, the overall analysis showed that there was no interaction between distractor condition and mood, $F(4, 1696) = 0.34$, $p = .850$, $\eta_p^2 < .01$, thereby confirming that mood did not affect auditory distraction. There was evidence of an auditory-deviant effect, $F(1, 848) = 44.98$, $p < .001$, $\eta_p^2 = .05$, but the auditory deviant effect did not differ as a function of mood, $F(2, 848) = 0.54$, $p = .585$, $\eta_p^2 < .01$. There was also evidence of a changing-state effect, $F(1, 848) = 276.96$, $p < .001$, $\eta_p^2 = .25$, but the changing-state effect did not differ as a function of mood either, $F(2, 848) = 0.28$, $p = .754$, $\eta_p^2 < .01$. Consequently, the results of the current study indicate that both the size of the auditory-deviant effect and the size of the changing-state effect remain stable in positive and negative moods compared to neutral ones.

The evidence of an effect of mood on auditory distraction in cross-modal paradigms is inconsistent [e.g., 37–39]. While the results of the current study are in line with the results of Hoskin et al. [39] who found no evidence of an effect of emotional state on distraction by auditory deviants, our results at first glance seem to differ from the results of Pacheco-Unguetti and Parmentier [37, 38] who found that happy and sad mood compared to neutral mood amplified the effects of deviant distractors in a cross-modal oddball paradigm. However, it is important to note that deviance distraction was defined as an increase in response latencies in an odd-even digit categorization task in those previous studies [37, 38]. In the present study, in contrast, distraction was measured in terms of the decrease in the proportion of correct answers in the serial-recall task. Hence, it is possible that differences in mood state may primarily affect the speed of responses while accuracy remains unaffected. Additionally, the difference in task difficulty needs to be taken into account. While we found no influence of mood on auditory distraction in a cognitively demanding serial-recall task in the current study, we cannot draw conclusions about how mood will influence auditory distraction in less demanding tasks [37, 38] because task difficulty may affect auditory distraction [74].

In the present study, we examined auditory distraction by assessing the auditory-deviant effect and the changing-state effect [e.g., 12, 16], that is, by measuring task performance in situations in which some kind of distraction is always present. This was done because the duplex-mechanism account [22] and the automatic-capture account [e.g., 8] provide clear predictions as to how mood should influence these types of auditory distraction. As a consequence, we cannot draw conclusions on how auditory distraction will be influenced when a quiet condition is also included in which case no distraction is present in some trials. Neither the duplex-mechanism account nor the automatic-capture account [8, 22] allow deriving clear predictions as to how a quiet condition would influence the extent of auditory distraction and the influence of mood on auditory distraction. This is nevertheless an interesting aspect for future research for the following reasons: First, quiet, distraction-free phases often occur in everyday situations in which one cannot necessarily predict whether or when auditory distraction might appear. Second, recent evidence suggests that—contrary to previous assumptions [e.g., 10, 22] —there is noteworthy disruption of serial recall when comparing a steady-state condition (used here as a baseline for determining the auditory-deviant and the changing-state effect) with a quiet condition [75, 76].

It also seems important to note that the emotional mood states that were manipulated in the present study were not directly associated with the serial-recall task because the aim was to draw conclusions about effects of persistent mood states on auditory distraction. This focus differs from that of other studies in which the task is to attend to emotional stimuli [e.g., 77]. Furthermore, the present results do not directly speak to the role of performance-related subjective states that are more directly linked to the task at hand such as task engagement or the

lack thereof (e.g., boredom). While the changing-state effect has consistently been found to be independent of factors that are likely to affect task engagement [e.g., 8, 24], inconsistent effects have been observed with regard to the auditory-deviant effect. Several studies have demonstrated that the auditory-deviant effect is abolished when the visual targets are masked by visual noise—which was interpreted as an effect of a compensatory increase in task engagement [24, 64]—, but recently it has been shown that monetary incentives increase task engagement but do not affect either the changing-state effect or the auditory-deviant effect [8]. Together with the evidence against a modulation of auditory distraction by task motivation [8], the present results may indicate that effects of auditory distraction on serial recall are persistent stimulus-driven processes that remain remarkably unaffected by subjective states. However, given the mixed evidence so far, more systematic examinations of the influence of situational factors such as time pressure, boredom, fatigue as well as a careful assessment of the accompanying subjective states are desirable to reach robust conclusions.

In conclusion, a confrontation with auditory stimuli during an unrelated task interferes with the focus on that task and impairs performance. Apart from their relevance for theories of auditory distraction, the present results are also interesting from an applied point of view as they suggest that auditory distraction is a pervasive problem in emotionally loaded situations and that, regardless of the mood people are in, it remains a challenge to stay focused in a distraction-filled world. It is thus important to protect performance from distraction in work and educational settings where accurate performance is required.

## Acknowledgments

We acknowledge support by the Heinrich Heine University Düsseldorf.

## Author Contributions

**Conceptualization:** Saskia Kaiser, Axel Buchner, Raoul Bell.

**Formal analysis:** Saskia Kaiser, Axel Buchner, Raoul Bell.

**Investigation:** Saskia Kaiser.

**Methodology:** Saskia Kaiser, Axel Buchner, Raoul Bell.

**Project administration:** Saskia Kaiser.

**Supervision:** Axel Buchner, Raoul Bell.

**Visualization:** Saskia Kaiser, Raoul Bell.

**Writing – original draft:** Saskia Kaiser.

**Writing – review & editing:** Axel Buchner, Raoul Bell.

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
