## [Decision Letter · Decision Letter 0]

8 Sep 2021

PONE-D-21-08795Positive and negative mood states do not influence cross-modal auditory distraction in the serial-recall paradigmPLOS ONE

Dear Dr. Kaiser,

Thank you for submitting your manuscript to PLOS ONE. I have now received the comments from two experts, who were mainly positive about your manuscript. There are, however, a few issues that you should consider before publication, which are mainly related to the mood induction procedure (between-subjects).  I invite you to submit a revised version of the manuscript that addresses the points raised during the review process.

We look forward to receiving your revised manuscript.

Kind regards,

José A Hinojosa, Ph.D.

Academic Editor

PLOS ONE

Journal Requirements:

Reviewers' comments:

Reviewer's Responses to Questions

**Comments to the Author**

1. Is the manuscript technically sound, and do the data support the conclusions?

Reviewer #1: Yes

Reviewer #2: Yes

2. Has the statistical analysis been performed appropriately and rigorously? 

Reviewer #1: Yes

Reviewer #2: Yes

3. Have the authors made all data underlying the findings in their manuscript fully available?

Reviewer #1: Yes

Reviewer #2: Yes

4. Is the manuscript presented in an intelligible fashion and written in standard English?

Reviewer #1: Yes

Reviewer #2: Yes

5. Review Comments to the Author

Reviewer #1: The authors examined the potential for a person’s mood to influence the size of their distraction from auditory distractors in a serial recall paradigm. The logic of the series of experiments was clearly spelled out in the introduction, and the clear writing continued throughout the manuscript. The experiments were well designed, included appropriate sample sizes to detect the relevant effect(s), and the data are available via OSF. The authors are to be commended for their excellent work.

In the literature on serial recall in the presence of auditory distractions, there are two main views: a unitary attentional view and a duplex-mechanism view. The authors took care to include both auditory deviant and changing-state conditions and contrasted these with the steady-state sounds condition to determine the size of the relative distraction effects. To distinguish between the unitary and duplex-mechanism views, it is important to have both “effects” in the same sample.

Through four experiments, mood induction techniques were adjusted and modified to ensure that mood was, in fact, changed by the manipulations. Despite clear effects of the mood induction and despite 4 experiments to self-replicate and to include both positive and negative mood states, there were no interactions between mood state and size of the auditory distraction effects.

Can the authors speculate on how the presence of a silence or quiet condition would influence the results? Perhaps having some type of sound on every trial led to a different level of resistance to any mood x distractor interaction, as compared to examining any type of sound relative to a quiet condition and the potential interaction with mood state. Indeed, the expectation becomes “there is always a distracting sound” and might that differ from the expectation of “there is sometimes a distracting sound”? Prior research has demonstrated that expectations are important in determining how irrelevant auditory stimuli are processed. This question of which auditory conditions to include or to exclude seems to be an area of the literature that has not been explored in enough depth and may help to clarify the differences between the unitary and duplex-mechanism views in future research.

I support the publication of this strong manuscript with minor revisions.

I noticed only minor typos:

Page 3. Line 36 “exciting” and not exiting

Page 26 line 610 was powerful “enough”? The word “enough” seems to be missing.

Reviewer #2: This interesting study examines the effects of mood induction (positive/negative, versus neutral) on auditory distraction. They use two mechanisms of distraction: “changing-state effect”, produced by sequences of different distractors over time. The subject processes more the order of these distractors, instead of the order of the targets required by the task, and this causes interference (i.e. interference-by-process). And an “auditory-deviant effect”, produced by deviant distractors.

In 4 behavioral experiments, mood induction is done with either autobiographical recall and music, or emotional pictures from IAPS. The authors conclude that mood does not affect auditory distraction in any case.

The paper is well-written and clear (but see below), with clear and interesting conclusions. The methodology seems sound for the posed questions, although I have the following comments/questions:

-The large sample size is appreciated (~200 per experiment). However, how many trials per condition were analyzed in each experiment? Was it 8 trials per distraction condition? (page 11) This number seems low for a reliable interpretation of the results. Please justify.

-It seems like the subjects were all different individuals from experiment to experiment. Was this the case? Please clarify when presenting experiments 2, 3 and 4.

-About the serial recall paradigm, it is said that “The immediate recall of information is severely disrupted when auditory distractors have to be ignored.” However, if the task is sufficiently difficult, auditory distraction may also be reduced (e.g. SanMiguel et al., 2008). One may wonder whether a task with a certain difficulty, such as this one, may hinder any effects of mood state, or any other contextual effects, on distraction. This could be briefly mentioned.

-In page 5, it is said: “…distraction by changing-state sequences is thought to arise from the obligatory processing of the auditory distractors, the changing state effect is assumed to be largely independent of the individual’s emotional-motivational state.” Then: “the auditory-deviant effect is the result of a trade off between the deployment of attention to the visual primary task and the allocation of attention to the task-irrelevant modality. Consequently, the auditory-deviant effect should be directly influenced by emotional-motivational factors that modulate this trade-off.” From this explanation and the text below in the same page (or the Discussion), I don’t see why one type of distraction should be more “obligatory” than the other, or “more independent of mood” than the other. Please explain in different words to clarify.

-Mood induction (positive/neutral in exp. 1-3, or sad-negative/neutral in exp. 2-4) was implemented in 2 different groups of participants. This is, in my opinion, a limitation, as the mood effects (or lack thereof) reported are actually group effects, and this could always be susceptible to confounds, even if mood induction worked as expected in either group. For instance, were individual anxiety levels similar across groups? But other group factors could also be affecting the results. In fact, in experiment 1, the positive mood group had a more positive mood from baseline, compared to the neutral group, which may have caused some sort of ceiling effect of mood on the subsequent distraction paradigm. It would have been perhaps more suitable to test the same sample of participants with both the control (neutral) and the emotional condition, perhaps in 2 different days, or performing the serial recall task both before and after mood induction in the same session (by e.g. counterbalancing the order of neutral/emotional mood inductions across subjects, to control for possible learning effects of the task). This could be at least acknowledged in the discussion.

-Another limitation of experiments 1 and 2 is the fact that mood induction is only implemented at the beginning of each experiment, and mood effects could have faded out over the 30 mins of duration of the whole session. In fact, in experiment 2 there seems to be decrease in negative affect over time in the sad-mood group. Also, the arousal effects (which could play a role on the effects on distraction) may have also disappeared, even if mood remains. The authors perform the next 2 experiments to overcome this limitation by using pictures, and this time interspersing the emotional pictures in the distraction task, which seems more appropriate. But this is a different way of inducing mood. So, in experiments 1, 2 and 4, did mood x distraction interactions remain invariable (non-significant) over time? The effects of mood on distraction could have appeared only at the beginning of these experiments (even when no apparent changes in mood are observed over time). This is only formally tested in experiment 3 (with no effects of trial order). This is relevant, especially for experiments 1 and 2, as their results contradict those in Pacheco-Unguetti & Parmentier (cited in the text).

-In experiments 3 and 4, pictures and distraction trials were separated in time by about 20 seconds. This is long comparatively to other studies showing that distractors and emotional stimuli had to be close in time for emotion effects to occur on the processing of the distractors (e.g. Selinger et al., 2008, Dominguez-Borras et al., 2017). To my knowledge these studies did not address mood induction directly, but delay of the distractor onsets relative to emotional stimuli might be affecting the results.

Minor:

-Page 5, line 94: there is a typo in “stimulus-aspecific”.

-Page 15, line 323: “..which correspondents to…” should be “which corresponds to…”

-Significant effects should be depicted in the Figures (bar graphs).

References

Domínguez-Borràs J, Rieger SW, Corradi-Dell'Acqua C, Neveu R, Vuilleumier P. Fear Spreading Across Senses: Visual Emotional Events Alter Cortical Responses to Touch, Audition, and Vision. Cereb Cortex. 2017 Jan 1;27(1):68-82.

SanMiguel I, Corral MJ, Escera C. When loading working memory reduces distraction: behavioral and electrophysiological evidence from an auditory-visual distraction paradigm. J Cogn Neurosci. 2008 Jul;20(7):1131-45.

Selinger L, Domínguez-Borràs J, Escera C. Phasic boosting of auditory perception by visual emotion. Biol Psychol. 2013 Dec;94(3):471-8.

6. PLOS authors have the option to publish the peer review history of their article (what does this mean?). If published, this will include your full peer review and any attached files.

Reviewer #1: **Yes: **Emily M Elliott

Reviewer #2: No

---

## [Author Response · Author response to Decision Letter 0]

19 Oct 2021

All points raised by the reviewers and regarding style requirements of PLOS ONE are addressed in the files named "Cover Letter" and "Response to Reviewers".

The answers to the points raised by the reviewers, as provided in the file named "Response to reviewers", are also presented below:

Response to the Reviewers: Manuscript “Positive and negative mood states do not influence cross-modal auditory distraction in the serial-recall paradigm”

Dear Dr. Hinojosa,

thank you for inviting a revision and for the helpful comments on our manuscript “Positive and negative mood states do not influence cross-modal auditory distraction in the serial-recall paradigm”. We have addressed all comments raised by the reviewers as described below.

For clarification, we would also like to point out that we corrected a few additional typos that we identified in the manuscript, included a correction note to Pacheco-Unguetti and Parmentier (2014) in the reference list and made minor changes to meet all style requirements of PLOS ONE and to clarify the wording throughout the manuscript. Information about these changes are also mentioned in the cover letter.

We highlighted all changes made to the manuscript and the reference list in red in the file named “Revised Manuscript with Track Changes”.

As before, the data referred to in the submitted manuscript as well as the supplementary material will be available from the project page at the Open Science Framework via https://osf.io/tqjwr/. Prior to publication, the data and the supplementary material can be accessed via this

view-only link: https://osf.io/tqjwr/?view_only=282f3b793a154bbf93fc09dd95b18202. In the original submission we provided both links, while in the resubmitted manuscript we only provide the link which should be used after publication. However, data are still accessible via the view-only link until the publication of the manuscript.

In the following, we would like to address the points raised by the reviewers. In our answers, we refer to the pages and lines of the manuscript where we made changes in order to accommodate the reviewers suggestions. The line numbers correspond to the relevant lines in

the file named “Revised Manuscript with Track Changes”.

Please do not hesitate to contact me in case you have any questions.

Responses to the points raised by Reviewer #1:

1. The authors examined the potential for a person’s mood to influence the size of their distraction from auditory distractors in a serial recall paradigm. The logic of the series of experiments was clearly spelled out in the introduction, and the clear writing continued throughout the manuscript. The experiments were well designed, included appropriate sample sizes to detect the relevant effect(s), and the data are available via OSF. The authors are to be commended for their excellent work.

Response: Thank you for the positive feedback.

2. Can the authors speculate on how the presence of a silence or quiet condition would influence the results? Perhaps having some type of sound on every trial led to a different level of resistance to any mood x distractor interaction, as compared to examining any type of sound relative to a quiet condition and the potential interaction with mood state. Indeed, the expectation becomes “there is always a distracting sound” and might that differ from the expectation of “there is sometimes a distracting sound”? Prior research has demonstrated that expectations are important in determining how irrelevant auditory stimuli are processed. This question of which auditory conditions to include or to exclude seems to be an area of the literature that has not been explored in enough depth and may help to clarify the differences between the unitary and duplex-mechanism views in future research.

I support the publication of this strong manuscript with minor revisions.

Response: Thank you for supporting the publication of our manuscript. In the present study, we included a steady-state condition as the control condition because the study was designed to test predictions of the duplex-mechanism account (Hughes, 2014) that makes strong predictions about the changing-state effect and the auditory-deviant effect, which are defined as a greater disruption by changing-state sequences and auditory-deviant sequences in comparison to steady-state sequences, respectively. The duplex-mechanism account does not make strong predictions about the inclusion of a silence or quiet condition. Nevertheless, we agree that systematically analyzing the effects of a quiet condition certainly is an interesting avenue for future research, particularly in light of recent evidence suggesting that—contrary to previous assumptions in the literature (Hughes, 2014)—there is noteworthy disruption of serial recall when comparing a steady-state condition to a quiet condition (AuBuchon et al., 2019; Bell et al., 2019). We have therefore added a paragraph in the General Discussion of the manuscript in which we discuss the relevance of a quiet condition for explanatory approaches regarding auditory distraction (page 31, lines 740-754).

3. I noticed only minor typos:

• Page 3. Line 36 “exciting” and not exiting

• Page 26 line 610 was powerful “enough”? The word “enough” seems to be missing.

Response: Thank you for pointing that out. We have corrected these typos. In the revised manuscript the corrections can be found in lines 40 and 643, respectively.

Responses to the points raised by Reviewer #2:

The paper is well-written and clear (but see below), with clear and interesting conclusions. The methodology seems sound for the posed questions, although I have the following comments/questions:

1. The large sample size is appreciated (~200 per experiment). However, how many trials per condition were analyzed in each experiment? Was it 8 trials per distraction condition? (page 11) This number seems low for a reliable interpretation of the results. Please justify.

Response: There were indeed eight trials per distractor condition that were aggregated at a participant level. In each trial, eight digits were presented that had to be remembered. This is a typical number of trials per distractor condition in the serial-recall paradigm that has proven to be useful in this particular paradigm (Bell et al., 2021; Buchner et al., 2004; Hughes et al., 2013; Marsh et al., 2020; Röer et al., 2015). We now

clarify in the manuscript that we closely followed the standard procedures of the serial-recall paradigm (page 11, lines 257-259).

2. It seems like the subjects were all different individuals from experiment to experiment. Was this the case? Please clarify when presenting experiments 2, 3 and 4.

Response: Thank you for pointing out this lack of clarity. We clarified this in the manuscript (page 16, lines 363-364; page 25, lines 578-579).

3. About the serial recall paradigm, it is said that “The immediate recall of information is severely disrupted when auditory distractors have to be ignored.” However, if the task is sufficiently difficult, auditory distraction may also be reduced (e.g. SanMiguel et al., 2008). One may wonder whether a task with a certain difficulty, such as this one, may hinder any effects of mood state, or any other contextual effects, on distraction. This could be briefly mentioned.

Response: Thank you for pointing this out. We added the consideration of task difficulty—including a citation of SanMiguel et al. (2008)—to the manuscript (page 31 lines 735-739).

4. In page 5, it is said: “…distraction by changing-state sequences is thought to arise from the obligatory processing of the auditory distractors, the changing state effect is assumed to be largely independent of the individual’s emotional-motivational state.” Then: “the auditory-deviant effect is the result of a trade off between the deployment of attention to the visual primary task and the allocation of attention to the task-irrelevant modality. Consequently, the auditory-deviant effect should be directly influenced by emotional-motivational factors that modulate this trade-off.” From this explanation and the text below in the same page (or the Discussion), I don’t see why one type of distraction should be more “obligatory” than

the other, or “more independent of mood” than the other. Please explain in different words to clarify.

Response: We absolutely share the reviewer’s view that attention can be voluntarily employed or involuntarily captured. However, this point in the manuscript refers to the assumptions and predictions of the duplexmechanism account (Hughes, 2014). Within the duplex-mechanism account, the distraction by auditory deviants is conceptualized as an attentional tradeoff that is influenced by voluntary task engagement and mood states. The distraction by changing-state sequences, by contrast, is conceptualized as passive interference, and thus postulated to be automatic and, hence, should not depend on mood states. To make this point clearer, we have carefully revised the manuscript to clarify the predictions of the duplex-mechanism account (pages 5-6, lines 89-114).

5. Mood induction (positive/neutral in exp. 1-3, or sad-negative/neutral in exp. 2-4) was implemented in 2 different groups of participants. This is, in my opinion, a limitation, as the mood effects (or lack thereof) reported are actually group effects, and this could always be susceptible to confounds, even if mood induction worked as expected in either group. For instance, were individual anxiety levels similar across groups? But other group factors could also be affecting the results. In fact, in experiment 1, the positive mood group had a more positive mood from baseline, compared to the neutral group, which may have caused some sort of ceiling effect of mood on the subsequent distraction paradigm. It would have been perhaps more suitable to test the same sample of participants with both the control (neutral) and the emotional condition, perhaps in 2 different days, or performing the serial recall task both before and after mood induction in the same session (by e.g. counterbalancing the order of neutral/emotional mood inductions across subjects, to control for possible learning effects of the task). This could be at least acknowledged in the discussion.

Response: To prevent confounds in the group design, we randomized the assignment of the participants to one of the two mood induction groups (positive / negative and neutral) by alternatingly assigning the participants to one group or the other solely based on the order of appearance in the lab (page 11, lines 246-248). Randomization is efficient to prevent confounding factors and enhance internal validity (Graziano & Raulin, 2010), especially when combined with large sample sizes. 

There was, indeed, an a-priori difference of positive affect between the positive and the neutral mood groups in Experiment 1. We mention in the manuscript (pages 12-13, lines 293-295) that this is most likely due to the fact that we informed the participants about the nature of the mood induction in the consent form so that they knew, at the time their mood was tested, that they would be recalling positive life events later on in

the experiment. It is easy to imagine that some participants may have already started thinking about positive life events, which may have already lifted their mood before the nominal mood-induction phase. As the data show, this does not affect the efficiency of the mood manipulation, but we do of course agree that this aspect of the procedure was not ideal for assessing baseline mood. Therefore, we altered the procedure for the following experiments. In consequence, no baseline mood differences were found in the following experiments (which suggests that the randomization procedure was successful). Furthermore, the more positive baseline of the positive mood group in Experiment 1 in unlikely to have caused a ceiling effect as positive affect was much higher after the positive mood induction than before the positive mood induction, providing evidence that the mood induction procedure was successful.

6. Another limitation of experiments 1 and 2 is the fact that mood induction is only implemented at the beginning of each experiment, and mood effects could have faded out over the 30 mins of duration of the whole session. In fact, in experiment 2 there seems to be decrease in negative affect over time in the sad-mood group. Also, the arousal effects (which could play a role on the effects on distraction) may have also disappeared, even if mood remains. The authors perform the next 2 experiments to overcome this limitation by using pictures, and this time interspersing the emotional pictures in the distraction task, which seems more appropriate. But this is a different way of inducing mood. So, in experiments 1, 2 and 4, did mood x distraction interactions remain invariable (non-significant) over time? The effects of mood on distraction could have appeared only at the beginning of these experiments (even when no apparent changes in mood are observed over time). This is only formally tested in experiment 3 (with no effects of trial order). This is relevant, especially for experiments 1 and 2, as their results contradict those in Pacheco-Unguetti & Parmentier (cited in the text).

Response: In Experiments 1 and 2, we used a standard mood induction procedure that has proven to be comparatively effective in the literature. However, it is already known from the literature that a common problem is that mood states fade out over time. To address this limitation of Experiments 1 and 2, we have performed Experiments 3 and 4 in which the mood induction was interspersed in between experimental trials. This procedure has proven to be effective as mood states persisted until after the serial-recall phase in Experiments 3 and 4. As suggested, we have now performed the additional analyses including time course for Experiments 1, 2 and 4. Time course did not significantly interact with any of the other variables, suggesting that changes in mood over the course of the experiment did not affect distraction. This information is now included in the General Discussion (pages 29-30, lines 691-704).

7. In experiments 3 and 4, pictures and distraction trials were separated in time by about 20 seconds. This is long comparatively to other studies showing that distractors and emotional stimuli had to be close in time for emotion effects to occur on the processing of the distractors (e.g. Selinger et al., 2008, Dominguez-Borras et al., 2017). To my knowledge these studies did not address mood induction directly, but delay of the distractor onsets relative to emotional stimuli might be affecting the results.

Response: This seems to be a misunderstanding. We have now carefully revised the manuscript to clarify the experimental mood induction procedure (pages 21-22, lines 492-501). Each pictures was presented—and focused on by the participants—for a total of 15 seconds. After five seconds of showing only the picture, the valence scale and the arousal scale of SAM were additionally presented at the bottom of the screen, but the emotional picture remained visible at the screen. Furthermore, participants were instructed to imagine themselves in the emotional scenes. After the 15 seconds had elapsed, a fixation cross was presented for one second, and then the distractor trial started. The emotional information thus was present until the start of each serial-recall trial. In these experiments we focused on the effect of persistent positive and negative moods states. While participants were in either a neutral, positive (Experiment 3) or negative (Experiment 4) mood during the serial recall trials, the mood and the mood induction were not directly coupled to the serial-recall task and hence the auditory distraction trials. This is a difference to the aforementioned papers (e.g., Domínguez-Borràs et al., 2017) as they presented auditory stimuli during the presentation of emotional pictures. This is now mentioned in the General Discussion (page 32, lines 755-758).

8. Minor:

• Page 5, line 94: there is a typo in “stimulus-aspecific”.

• Page 15, line 323: “..which correspondents to…” should be “which corresponds to…”

Response: Thank you for pointing that out. The correction can be found in line 344. The term “aspecific” was taken verbatim from Hughes (2014, p. 31). However, we agree that the term may be confusing without further information about why the effect is considered aspecific. Given that this distinction is irrelevant for the research question at hand, we have deleted this term.

• Significant effects should be depicted in the Figures (bar graphs).

Response: We seriously considered this suggestion. However, in the end we decided against depicting statistically significant effects in the figures because, given the many significant effects we report in the figures, depicting each significant effect would have caused visual clutter.

---

## [Decision Letter · Decision Letter 1]

16 Nov 2021

Positive and negative mood states do not influence cross-modal auditory distraction in the serial-recall paradigm

PONE-D-21-08795R1

Dear Dr. Kaiser,

We’re pleased to inform you that your manuscript has been judged scientifically suitable for publication and will be formally accepted for publication once it meets all outstanding technical requirements.

Kind regards,

José A Hinojosa, Ph.D.

Academic Editor

PLOS ONE

Additional Editor Comments (optional):

Reviewers' comments:

Reviewer's Responses to Questions

**Comments to the Author**

1. If the authors have adequately addressed your comments raised in a previous round of review and you feel that this manuscript is now acceptable for publication, you may indicate that here to bypass the “Comments to the Author” section, enter your conflict of interest statement in the “Confidential to Editor” section, and submit your "Accept" recommendation.

Reviewer #1: All comments have been addressed

Reviewer #2: All comments have been addressed

2. Is the manuscript technically sound, and do the data support the conclusions?

Reviewer #1: Yes

Reviewer #2: Yes

3. Has the statistical analysis been performed appropriately and rigorously? 

Reviewer #1: Yes

Reviewer #2: Yes

4. Have the authors made all data underlying the findings in their manuscript fully available?

Reviewer #1: Yes

Reviewer #2: Yes

5. Is the manuscript presented in an intelligible fashion and written in standard English?

Reviewer #1: Yes

Reviewer #2: Yes

6. Review Comments to the Author

Reviewer #1: (No Response)

Reviewer #2: (No Response)

7. PLOS authors have the option to publish the peer review history of their article (what does this mean?). If published, this will include your full peer review and any attached files.

Reviewer #1: No

Reviewer #2: No

---

## [Editor Report · Acceptance letter]

16 Dec 2021

PONE-D-21-08795R1 

Positive and negative mood states do not influence cross-modal auditory distraction in the serial-recall paradigm 

Dear Dr. Kaiser:

I'm pleased to inform you that your manuscript has been deemed suitable for publication in PLOS ONE. Congratulations! Your manuscript is now with our production department. 

Kind regards, 

on behalf of

Dr. José A Hinojosa 

Academic Editor

PLOS ONE